Genome-scale investigation of phenotypically distinct but nearly clonal Trichoderma strains

Lange Claudia 1 claudia.lange@lincolnuni.ac.nz
Weld Richard J. 1 2
Cox Murray P. 1 3
Bradshaw Rosie E. 1 3
McLean Kirstin L. 1
Stewart Alison 4
Steyaert Johanna M. 1
1 Bio-Protection Research Centre, Lincoln University , Lincoln , New Zealand
2 Lincoln Agritech Limited, Lincoln University , Lincoln , New Zealand
3 Institute of Fundamental Sciences, Massey University , Palmerston North , New Zealand
4 Forest Science, Scion , Rotorua , New Zealand
Newton Irene
Electronic publication date: 2016 May 12
Publication date: 2016
Volume: 4
Electronic Location ID: e2023
Received 2016 Mar 9; Accepted 2016 Apr 19
Copyright: ©2016 Lange et al.
Copyright year: 2016
Copyright holder: Lange et al.
License: This is an open access article distributed under the terms of the Creative Commons Attribution License, which permits unrestricted use, distribution, reproduction and adaptation in any medium and for any purpose provided that it is properly attributed. For attribution, the original author(s), title, publication source (PeerJ) and either DOI or URL of the article must be cited.
License URL: https://creativecommons.org/licenses/by/4.0/

Keywords: Discordant phenotypes, Biocontrol, Trichoderma cf. atroviride, Single nucleotide polymorphism (SNP), Genomics, Small ERDK-rich factor (serf), Molecular marker

Funding: Lincoln University Bio-Protection Research Centre, Lincoln, New Zealand This research was funded by Lincoln University and the Bio-Protection Research Centre, Lincoln, New Zealand. The funders had no role in study design, data collection and analysis, decision to publish, or preparation of the manuscript.

==============================
Biological control agents (BCA) are beneficial organisms that are applied to protect plants from pests. Many fungi of the genus Trichoderma are successful BCAs but the underlying mechanisms are not yet fully understood. Trichoderma cf. atroviride strain LU132 is a remarkably effective BCA compared to T. cf. atroviride strain LU140 but these strains were found to be highly similar at the DNA sequence level. This unusual combination of phenotypic variability and high DNA sequence similarity between separately isolated strains prompted us to undertake a genome comparison study in order to identify DNA polymorphisms. We further investigated if the polymorphisms had functional effects on the phenotypes. The two strains were clearly identified as individuals, exhibiting different growth rates, conidiation and metabolism. Superior pathogen control demonstrated by LU132 depended on its faster growth, which is a prerequisite for successful distribution and competition. Genome sequencing identified only one non-synonymous single nucleotide polymorphism (SNP) between the strains. Based on this SNP, we successfully designed and validated an RFLP protocol that can be used to differentiate LU132 from LU140 and other Trichoderma strains. This SNP changed the amino acid sequence of SERF, encoded by the previously undescribed single copy gene “small EDRK-rich factor” (serf). A deletion of serf in the two strains did not lead to identical phenotypes, suggesting that, in addition to the single functional SNP between the nearly clonal Trichoderma cf. atroviride strains, other non-genomic factors contribute to their phenotypic variation. This finding is significant as it shows that genomics is an extremely useful but not exhaustive tool for the study of biocontrol complexity and for strain typing.

Introduction

The filamentous fungi Trichoderma atroviride, and the closely related Trichoderma cf. atroviride (M Braithwaite, PR Johnston, KL McLean, F Nourozi, AJ Hay, M Ohkura, C Lange, P Shoukouhi, RA Hill, S Ball, NJ Cummings, D Bienkowski, A Stewart, JM Steyaert & J Bissett, 2015, unpublished data), are species of economic and ecological importance to horticulture and agriculture for biological control of a variety of plant pathogens. However, there is considerable strain-specific variation in biocontrol characteristics, exemplified by T. cf. atroviride biocontrol strains LU132 (formerly C52) and LU140 (formerly D73). These strains were isolated three years apart from the same New Zealand onion field but extensive trials revealed differences in their biocontrol abilities with pathogenic fungi (Dodd-Wilson, 1996; Harrison & Stewart, 1988; Kay, 1991; Kay, 1994; Kay & Stewart, 1994; Lange, 2015; McLean, 1996; McLean, 2001). The two strains are propagated asexually by mycelial growth and the production of conidia. Both strains were found to be antagonistic towards the onion pathogen Sclerotium cepivorum (Harrison & Stewart, 1988; Kay & Stewart, 1994; McLean, 1996). As they consistently grew more rapidly than S. cepivorum and showed no evidence of antibiotic production or direct antagonism, their chief biocontrol mechanism was suggested to be competition for nutrients and space (Harrison & Stewart, 1988; Kay, 1994). However, LU132 was substantially more efficient than LU140 in inhibiting S. cepivorum on agar plates and in the glasshouse (Kay, 1994; McLean, 1996; McLean, 2001), as well as in proliferation and establishment in soil (McLean, 1996), promoting onion seedling emergence (McLean & Stewart, 2000) and growth promotion of rye grass (Chohan et al., 2010). Due to the excellent biocontrol of S. cepivorum and Botrytis cinerea demonstrated by LU132, it was formulated into the commercial biocontrol products Trichopel® Ali 52, Sentinel® and Tenet® (Agrimm Technologies Ltd., Lincoln, New Zealand) (Card et al., 2009; McLean et al., 2012; McLean et al., 2005).

In order to release a new biocontrol product on the European market, authorities require a molecular identification method to monitor the development and distribution of the organism in the environment. Attempts were made to develop an LU132-specific molecular marker so its ecology could be monitored within commercial systems. Markers are commonly designed on the basis of individual fingerprinting profiles. However, in the case of LU132, this approach was not sufficient, as the profiles could distinguish LU132 from all tested strains except the less effective strain LU140. Fingerprinting profiles, a SCAR approach (Cordier et al., 2007; Hermosa et al., 2001; Naeimi et al., 2011) and sequencing of a number of marker genes and regions (the translation elongation factor 1α (tef1 α) (Shoukouhi & Bissett, 2009), the 42 kDa endochitinase (ech42) (Lieckfeldt et al., 2000), the Internal Transcribed Spacer regions 1 and 2 (ITS1 and ITS2) of rRNA genes (Kuhls et al., 1997; Rehner & Samuels, 1994) and the mitochondrial cytochrome oxidase subunit 1 encoding gene cox1 (Hamari et al., 2003)) could not identify any DNA sequence differences between LU132 and LU140 (Dodd-Wilson, 1996; McLean, 2001 and unpublished data from within our research group). Therefore a genome-wide search for polymorphisms was required.

Much genetic research has focussed on fungi as eukaryotic model organisms due to their relatively small genome sizes and simplistic lifestyles (Van der Klei & Veenhuis, 2006). However, many fungi have interesting traits in their own right. The genus Trichoderma comprises more than 200 species (Atanasova, Druzhinina & Jaklitsch, 2013), of which many form endophytic relationships with higher plants (Harman et al., 2004). This close relationship promotes plant growth and activates the plant’s resistance to pests. In addition, they parasitise plant pathogens (mycoparasitism) and produce bioactive secondary metabolites and enzymes, which makes them valuable in agriculture as BCA and in industry as sources of hydrolytic enzymes (Harman et al., 2012; Harman et al., 2004; Lorito et al., 2010).

To understand what makes a good Trichoderma BCA, many studies have focussed on mycoparasitism related genes (e.g., cell wall degrading enzymes) and genes that promote plant growth or induce systemic resistance in the plants (Harman et al., 2012). A recent genome comparison between one hypercellulolytic species (Trichoderma reesei) and two biocontrol species (T. atroviride and T. virens) found that the two biocontrol species contained more mycoparasitism-relevant genes, such as genes coding for chitinolytic enzymes, antibiotics and toxins, than the hypercellulolytic species (Kubicek et al., 2011). However, the complexity of biocontrol interactions between the pathogen, plant and antagonist makes it difficult to link the phenotype of a good BCA to a specific genetic origin. The high genetic similarity, but phenotypic differences, between T. cf. atroviride LU132 and LU140 presented a rare opportunity to compare nearly identical biocontrol fungi in order to study biocontrol-specific gene variants.

The aim of this research was to search for genetic differences (SNPs) between LU132 and LU140 on a whole genome scale to enable a strain-specific DNA-based distinction. We further investigated whether the genetic differences influence the distinct phenotypes. Initially, LU132 and LU140 phenotypes were compared directly to each other to confirm their individuality, to describe and quantify the different characteristics and to predict what target genes might be important. Then, the whole genomes of LU132 and LU140 were sequenced to identify SNPs and an LU132-specific molecular marker was designed. Target genes whose function might be affected by the SNPs were identified and the effects of changes in target genes on the phenotypes of LU132 and LU140 were examined by gene deletion experiments.

Materials and Methods

Fungal strains

We refer to LU132 and LU140 as T. cf. atroviride because a recent five gene phylogeny of Trichoderma spp. from New Zealand (Tef1α, ACLA1, Calm1, LAS1 and RPB2) resulted in the definition of this new species, closely related to T. atroviride (M Braithwaite, PR Johnston, KL McLean, F Nourozi, AJ Hay, M Ohkura, C Lange, P Shoukouhi, RA Hill, S Ball, NJ Cummings, D Bienkowski, A Stewart, JM Steyaert & J Bissett, 2015, unpublished data).

Trichoderma atroviride strain IMI206040 was obtained from Alfredo Herrera-Estrella, (Langebio, Mexico) and T. cf. atroviride strains LU132 and LU140 from the Lincoln University Culture Collection (LUCC, New Zealand), where single spore isolations had been stored at −80 °C since the original strain isolations. Trichoderma inocula were prepared by growing the strains on Potato Dextrose Agar (PDA; Difco) at 25 °C with constant light for 8 d. The conidia were then suspended in water, filtered through two layers of Miracloth (Calbiochem®); the conidial suspension was adjusted to 1 × 109 conidia per mL and aliquots stored in 25% glycerol at −80 °C. Fresh cultures of Botrytis cinerea (BC106) and Sclerotium cepivorum (LU360) were obtained from the LUCC, sub-cultured onto PDA and incubated at 20 °C with constant light for 14 d. The resulting sclerotia were stored in 25% glycerol at −80 °C. Agar plugs of Pythium ultimum spp. were sub-cultured onto PDA containing ampicillin, chloramphenicol and streptomycin (50, 100 and 50 µg/mL respectively) and incubated at 20 °C in complete darkness. Agar plugs were stored in 25% glycerol at 4 °C.

Growth rate and conidiation

Radial mycelial growth rates and colony morphologies of LU132 and LU140 were determined on standard 90 mm Petri dishes containing either buffered or un-buffered media in the pH range 2.4–6.0. The media were PDA (Difco); Malt Extract Agar (MEA, 30 g Malt extract (Difco), 5 g Peptone (Difco) 15 g agar per L water); or Minimal Medium agar with 0.2% glucose (MMA, (Carsolio et al., 1994; Steyaert et al., 2004)). Where indicated, the pH was adjusted as described previously (McIlvaine, 1921; Steyaert, Weld & Stewart, 2010) (Table S1). Agar plates were inoculated centrally with 2 µL conidial suspensions or with agar plugs containing one colony derived from a single conidium. Plates were incubated at 25 °C in total darkness or with constant light for up to 7 d. Conidial yield was determined for one treatment in Exp. 3 (Holder & Keyhani, 2005). The experimental design is outlined in Table S1.

Dual culture assay with plant pathogens

The plant pathogens Sclerotium cepivorum (ascomycete), Botrytis cinerea (ascomycete) and Pythium ultimum (oomycete) were selected to study the antagonistic activity of LU132 and LU140 on dual culture plates using methods described elsewhere (McLean, 1996). Five replicate plates were inoculated with 5 mm mycelial plugs from 3 d old cultures of the pathogens and Trichoderma test strains, 60 mm apart from each other.

Metabolic profiling

Phenotype MicroArrays™ for Filamentous Fungi (Biolog FF, Biolog Inc., Hayward, CA) were used to compare metabolic profiles of LU132 and LU140, utilising 95 single compounds. Assimilation of compounds was reflected by mycelial growth and quantified by measuring the optical density (OD) in the wells at 750 nm, the wavelength at which hyaline mycelium has its maximum absorbance. To quantify catabolism, the wells contain a tetrazolium dye that turns into a purple insoluble precipitate when reduced due to mitochondrial activity. This was measured at the maximum absorbance of the reduced tetrazolium salt of 490 nm (Bochner, Gadzinski & Panomitros, 2001). Conidia of LU132 and LU140 were generated on PDA plates at 25 °C in constant light. The FF plate procedure was carried out essentially as per manufacturer’s instructions and as described elsewhere (Bochner, Gadzinski & Panomitros, 2001) with modifications. Three replicate plates with separately prepared inocula were incubated at 25 °C under constant light. Cluster analysis was carried out as described elsewhere (Druzhinina et al., 2006). Absorbance readings over 2.0 and wells that contained conidia were excluded from the absorbance analysis as they would skew the data. In addition to absorbance measurements, conidiation was assessed using a scoring system from 0 to 5 (Friedl, Kubicek & Druzhinina, 2008).

Genome sequencing

Genomic DNA (gDNA) of LU132 and LU140 were prepared from mycelia using the Gentra® Puregene™ Tissue Kit (Qiagen) and further purified using the DNeasy® Plant Mini Kit (Qiagen) as per manufacturers’ instructions. DNA samples were sequenced with an Illumina GAII (Solexa) machine by the Massey University Genome Service, New Zealand. The data were analysed to identify Single Nucleotide Polymorphisms (SNPs) by mapping LU132 and LU140 reads against the unmasked genome sequence of the closest available reference genome of T. atroviride strain IMI206040 (Genome build v. 2.0, May 2010, Joint Genome Institute, http://genome.jgi.doe.gov/Triat2/Triat2.home.html) using the Burrows-Wheeler transform (BWT) algorithm implemented in the program bwa v.0.5.5 (Cox, Peterson & Biggs, 2010; Li & Durbin, 2009). Polymorphisms were identified using SAMtools (Li et al., 2009) and custom in-house software. At least 8 reads confirming a mutant allele were required to call a SNP at any given position. Unmapped reads were assembled de novo using ABySS (version 1.2.0) and Phrap (version 1.090518) and reads of one strain were mapped against the other’s assembled contigs (bwa version 0.5.8).

SNP confirmation

Regions that encompassed putative SNPs were PCR-amplified from gDNA of IMI206040, LU132 and LU140, then sequenced (primers are shown in Table S2). All PCR amplifications were performed in a Bio-Rad Icycler™ (Bio-Rad Laboratories) using the Expand long template PCR system (Roche) according to manufacturer’s instructions. SNP-containing genes (target genes) were identified using the reference genome annotation (IMI206040). The SNP-containing sequences were subjected to BLAST analysis on GenBank® to identify putative gene homologs in other organisms and possible biological functions.

Protein structure predictions

Deduced amino acid sequences that were altered as a consequence of a SNP were analysed to predict protein structures functional motifs using PSIPRED v3.3 (http://bioinf.cs.ucl.ac.uk/psipred/), SignalP 4.1 (http://www.cbs.dtu.dk/services/SignalP/), TargetP 1.1 (http://www.cbs.dtu.dk/services/TargetP/), NetSurfP 1.1 (http://www.cbs.dtu.dk/services/NetSurfP/), Phyre2 (http://www.sbg.bio.ic.ac.uk/servers/phyre2/html/page.cgi?id=index), SAM-T08 (http://compbio.soe.ucsc.edu/SAM_T08/T08-query.html) and ELM (http://elm.eu.org/search/).

Development and validation of a strain-specific molecular marker for LU132

The molecular test for the T. cf. atroviride biocontrol strain LU132 was developed based on PCR amplification of the genomic region containing an LU132-specific SNP, followed by restriction fragment length polymorphism (RFLP) analysis of the amplified fragment. The PCR amplification from gDNA of LU132, LU140, IMI206040 and 39 other T. atroviride and T. cf. atroviride strains from New Zealand, Europe and Asia was carried out as described above. The sequences from LU132, LU140 and IMI206040 were subjected to in silico restriction analysis using DNAMAN (v. 4.0a; Lynnon Corporation, Quebec, Canada) to identify SNP-specific restriction sites. The PCR fragments were digested to completion with the identified enzyme according to manufacturer’s instructions. Digested fragments were size fractionated by 2% agarose TAE gel electrophoresis for confirmation of strain-specific banding patterns.

Relative expression

The expression of identified target genes, relative to the expression of the reference gene encoding translation elongation factor 1α (tef1α) (Bustin et al., 2009; Seidl, Druzhinina & Kubicek, 2006), was studied under standard and specific inducing conditions. The standard culturing conditions were as follows: 100 mL PDB (Difco) were inoculated with 5 µL of LU132 and LU140 conidial suspensions in 250 mL conical flasks. The cultures were incubated at 25 °C in constant light with shaking at 200 rpm for 3 d. To induce a mycoparasitic response, the pH of the PDB was adjusted to 4.75 with HCl (Moreno-Mateos et al., 2007) and 4 h before the end of incubation, N-acetyl-D-glucosamine (NAG) (Sigma-Aldrich) was added to the culture to a final concentration of 0.5% (Mach et al., 1999). NAG is known to trigger the expression of target genes nag1 and nag2 and to induce biocontrol mechanisms (Brunner et al., 2003; Mach et al., 1999; Peterbauer et al., 1996; Ramot et al., 2004; Zeilinger et al., 1999). Mycelia from three replicates were harvested by filtration and snap frozen in liquid nitrogen. Total RNA was prepared using the Plant Total RNA Extraction Miniprep System (Viogene BioTek Corp.) and the RNA samples were treated with DNase using Turbo DNA-free™Kit (Ambion), as per manufacturers’ instructions. Intron-spanning primers were designed to amplify around 100 bp of transcript sequence (Table S2). Reverse Transcription quantitative real-time PCR amplification (RT-qPCR) and cycling conditions were based on the protocol from Holyoake et al. (2008), except using 10 ng RNA as template. All PCR reactions were done in duplicate and the whole experiment was repeated. The normalised gene expression data were expressed as ΔCq = Cq(tg) − Cq (tef1α) (Bustin et al., 2009).

Gene transcript sequencing

The mRNA transcripts of SNP-harbouring target genes, obtained from total RNA prepared as described above, were reverse transcribed into complementary DNA (cDNA) and sequenced using primers described in Table S2. First strand cDNA synthesis was carried out using SuperScript™ III Reverse Transcriptase (Invitrogen), as per manufacturer’s instructions, using the 3’UTR (or reverse) primers. Subsequent PCR amplifications were performed as above.

Mutational analyses

Primers used for the construct and vector creation are listed in Table S2. The serf knock-out construct (SKO) contained the hygromycin B phosphotransferase gene (hph) under the control of the pyruvate kinase gene promoter (pki) (Mach, Schindler & Kubicek, 1994), embedded in approximately 1 kb of each of the genomic 5′ and 3′ flanking sequences of serf. The knockout construct was then cloned into the binary vector pYT6 to create the new vector pSKO. The pSKO plasmid was electroporated to A. tumefaciens EHA105 cells using a MicroPulser™ Electroporator (Bio-Rad Laboratories) as per manufacturer’s instructions. Transformation of LU132 and LU140 was based on standard protocols (De Groot et al., 1998; Zwiers & De Waard, 2001) with modifications. pSKO-containing A. tumefaciens were selectively grown in media with 25 µg/mL kanamycin and 25 µg/ml rifampicin. Co-cultivation of 3 d old Trichoderma conidia and Agrobacterium colonies was carried out on sterile cellophane discs on IMAS agar, without overlay, at 23 °C. The cellophane disks were cut into three pieces, transferred onto separate PDA plates (Difco), containing 200 µg/mL hygromycin B and 300 µg/mL timentin (GlaxoSmithKline Plc.), and incubated at 25 °C for up to 4 d in complete darkness. Transformants were transferred to fresh PDA (Difco) containing antibiotics as above and purified via single-spore isolation. Homologous recombination at the serf locus was confirmed by PCR (Table S2) and Southern hybridisation (Sambrook, Fritsch & Maniatis, 1989). Mutants were characterised for growth, conidiation and metabolic activity as described above.

Statistical analyses

All data were analysed using GenStat (v. 14, VSN International Ltd., Hemel Hempstead, UK). Unless mentioned otherwise, data were analysed using the General Analysis of Variance method. The least significant differences of means (l.s.d.) and multiple comparisons (using Fisher’s Unprotected LSD algorithm) were determined at a significance level of 5% (P < 0.05).

Results

Phenotype comparison

Morphological analysis

Morphological comparisons were done to assess the extent of phenotypic differences between the T. cf. atroviride strains LU132 and LU140. LU132 grew consistently significantly faster than strain LU140 in all treatments (P < 0.05) (Table S3). For example, in experiment 2 (PDA, pH 5, unbuffered, dark) the average growth rates for LU132 and LU140 were 24.8 ± 0.45 mm/d and 21.0 ± 0.61 mm/d. The pH range that resulted in the fastest growth was wider for LU132 than for LU140 (Table S3). Colonies incubated with constant light also differed in the distribution and density of conidia (Fig. 1); whereas LU140 produced conidia all over the plate on PDA, LU132 conidia were more concentrated at the edges. This distribution effect was most apparent on unbuffered PDA with the lowest pH (2.9), on which LU140 produced 3.6 times more conidia than LU132 (1.8 × 109 and 0.5 × 109 conidia per plate on average respectively, P < 0.05).

Figure 1 Colony appearance/ conidiation patterns of LU132 and LU140.

LU132 conidiated less in the middle of the colony and more around the edges while LU140 conidiated over the whole plate on PDA. The most obvious conidiation difference between LU132 and LU140 was found on pH 2.9 PDA, where LU132 only conidiated around the edge of the plate (ub, unbuffered).

Dual culture with plant pathogens

LU132 and LU140 showed significant differences in biocontrol activity against pathogenic fungi. LU132 inhibited the growth of the plant pathogens Botrytis cinerea, Sclerotium cepivorum and Pythium ultimum significantly better than LU140 (P < 0.05) (Table S4). As soon as the pathogen colonies met the T. cf. atroviride colonies, they stopped growing. All pathogens were completely overgrown by both T. cf. atroviride strains after 12 d. There were no interaction zones between pathogens and antagonists. On the pathogen-only control plates with S. cepivorum and B. cinerea, sclerotia were observed, whereas no sclerotia were produced on the dual culture plates with T. cf. atroviride.

Metabolic profiling

The phenotype microarray assay (Biolog FF) assessed the growth, metabolism and conidiation of LU132 and LU140 in the presence of 95 single compounds on 96 well plates. A cluster analysis was done to group compounds that were metabolised by each strain at high, medium or low rates; based on absorbance measurements at OD490 and OD750 (representing catabolism and mycelial growth respectively). The six compounds for which the LU132 and LU140 strains differed most in their growth and/or metabolism are shown in Fig. 2. The amount of conidia in the wells was scored from 0 to 5 (Friedl, Kubicek & Druzhinina, 2008). The average conidiation score on all 95 compounds was significantly bigger for LU140 than for LU132 at each time point studied (60, 84 and 108 h). LU140 also produced conidia in more wells than LU132 throughout the experiment (Table S5). Apart from α-D-glucose, the compounds that were differentially metabolised by LU132 and LU140 (Fig. 2) did not result in altered conidiation. On α-D-glucose LU140 had a conidial mat covering the whole well at 60 h and LU132 at 84 h. The biggest conidiation score difference of 3 was found on D-galactose, D-raffinose, α-methyl-D-galactoside and stachyose at the end of the experiment (108 h).

Figure 2 Compounds that were differentially metabolised by LU132 and LU140.

Values with ∗ are significantly different (P < 0.05) between LU132 and LU140 at the respective wavelengths and time points. Time points at which conidia were present in the wells were not analysed.

The metabolism of the glucoside salicin was significantly better in LU132; however, the salicin metabolism could not be associated with a specific pathway. N-acetyl-D-glucosamine (NAG) was the only compound that induced differential metabolism in LU132 and LU140 and that could be associated with pathways implicated in biocontrol (Brunner et al., 2003; Lopez-Mondejar et al., 2009; Seidl, Druzhinina & Kubicek, 2006; Zeilinger et al., 1999). Trichoderma spp. have two genes encoding N-acetyl-β-D-glucosaminidases, nag1 and nag2 (Mach et al., 1999; Peterbauer et al., 1996; Ramot et al., 2004), which were selected as target genes for further analyses.

The results clearly identified LU132 and LU140 as two individuals. The strains exhibited distinct growth, conidiation and metabolism. LU132’s better pathogen control could be attributed to its faster growth.

Genome comparison

Whole genome sequences of LU132 and LU140 were mapped to the closest available reference genome of T. atroviride IMI206040 (Genome build v. 2.0, May 2010, Joint Genome Institute, http://genome.jgi.doe.gov/Triat2/Triat2.home.html) to identify single nucleotide polymorphisms (SNPs). The size of the reference genome is 36.1 Mb. It has 29 contigs and 11,863 gene models. Given the coverage threshold (the SNP calling requirement of at least 8 reads per allele) a read coverage of 87% and 73.5% was achieved for LU132 and LU140 respectively, with an average of 35 and 15 reads per nucleotide position for LU132 and LU140 respectively. Both T. cf. atroviride strains were found to have a sequence divergence rate of 2.5% compared to the reference strain T. atroviride IMI206040.

All identified 659 putative SNPs were validated empirically. By visualising the sequencing reads and the reference genome in IGV (v.1.5.65, Broad Institute), false positives were detected when putative SNPs occurred at the end of a read where the sequencing accuracy was low and deletions or insertions were introduced incorrectly by the mapping program. Some putative SNPs occurred within microsatellites or homopolymers and were therefore unreliable. Sanger sequencing of the SNP-encompassing regions confirmed two SNPs, both transition polymorphisms: SNP1, specific to LU132 (IMI206040 v. 2.0, contig_21:1,174,857, C→ T) and SNP2, specific to LU140 (IMI206040 v. 2.0, contig_15:842,707, A→ G). The remaining 657 putative SNPs were false positives, and there were no other SNPs in non-coding regions.

De novo assembly of unmapped reads was done to determine whether LU132 and LU140 differed in genome content, such as insertions or deletions that could account for their phenotypic differences. The assembly revealed that both LU132 and LU140 contained DNA sequences that were not present in the reference genome but none were specific for one strain. The contigs containing assembled sequences that mapped to LU132/LU140, but not the reference genome, were small and either matched to the φ X174 genome (which is used as control DNA in the Illumina sequencing process) or had high similarities to human and bacterial genes, suggestive of contamination.

Development and validation of a strain-specific molecular marker for LU132

A PCR to amplify a 660 bp region, which encompassed the LU132-specific SNP1, was carried out with LU132, LU140, IMI206040 and 39 other T. atroviride and T. cf. atroviride strains from New Zealand, Europe and Asia. Sequence analysis of the PCR products from LU132, LU140 and IMI206040 showed that LU132 contained three HphI restriction sites while the products of LU140 and IMI206040 contained two. The additional HphI site in LU132 was conferred by the LU132-specific SNP1 (Table 1).

All PCR products were digested to completion with HphI (New England Biolabs). Figure 3 shows the banding patterns for LU132, LU140 and IMI206040. The SNP1-containing region could also be amplified from five T. atroviride strains from Europe and Asia and three T. atroviride strains from New Zealand but the digest banding pattern resembled that of LU140 and IMI206040. Of 34 tested T. cf. atroviride strains from New Zealand, five could not be amplified and the remaining showed the LU140 and IMI206040 banding pattern (see strain identities and RFLP results in Table S6). These results confirmed the strain-specificity of the developed LU132-specific molecular RFLP marker.

Linking phenotype and genotype

The high genetic similarity in combination with the phenotypical distinctness of LU132 and LU140 led us to investigate functional relationships between the SNPs and phenotypic characteristics.

Table 1 LU132-specific Hph I restriction site created by SNP1.

	Sequence (5′ → 3′)	
IMI204060	TAAAGGCGAAGGTAGAAGCGAAAAT	
LU132	TAAAGGTGAAGGTAGAAGCGAAAAT	
LU140	TAAAGGCGAAGGTAGAAGCGAAAAT	
Hph I restriction sequence	....GGTGANNNNNNNN/.......	
Notes.

SNP1 (underlined) generated a third HphI restriction site in the PCR fragment that was only present in LU132 (at 350 bp).

Table 2 Average radial growth rates (mm/d) of WT and Δserf mutants.

Strain	Actual pH 5.0a	Actual pH 2.7a	Grand meanb	
	Dark		Light		Light	
LU132 WT	21.77	a	21.04	a	16.39	a	19.73	a	
LU132 A	21.15	b	18.68	c	14.28	c	18.04	c	
LU132 B	21.27	b	20.20	b	15.90	b	19.12	b	
LU132 C	21.23	b	20.12	b	15.89	b	19.08	b	
LU140 WT	20.54	c	18.31	cd	13.27	de	17.37	d	
LU140 D	20.57	c	17.58	e	12.99	e	17.05	d	
LU140 E	20.62	c	17.91	de	13.39	d	17.31	d	
LU140 F	20.70	c	18.03	d	13.18	de	17.30	d	
l.s.d.c	0.219		0.404		0.337		0.337		
Notes.

Growth rate data for LU132 and LU140 wild types (WT) and Δserf mutants (A–F). Values are averages of two experiments with four replicates. Different letters within a column represent significantly different values (P < 0.05).

a The pH was adjusted to 5.0 and 2.4. Before inoculation, the actual pH of the plates was measured.

b Grand mean is the average of the three conditions.

c The grand mean l.s.d. was determined using a split-plot design.

Figure 3 Electrophoretic separation of HphI digested PCR fragments.

Lanes B, D and F represent the undigested 660 bp PCR fragments from LU132, LU140 and IMI204060, respectively. Lanes A, C and E represent the HphI digested PCR fragments for LU132, LU140 and IMI204060, respectively. The arrow indicates the 350 bp restriction fragment that was only found in LU132. Size standard was 1 Kb Plus DNA Ladder™ (Invitrogen) in lane G.

Target genes directly associated with SNPs

The loci with which the two SNPs were associated were determined by analysis of their positions in the reference genome. SNP1 was located in the coding region of a predicted gene (protein ID 306899: GenBank accession number XP_013945379.1) with the putative conserved domain 4F5 (pfam04419) and similarities to the predicted small EDRK-rich factor (serf) that is conserved in fungi, protozoa and animals. 4F5 protein family members are short proteins with unknown function (Marchler-Bauer et al., 2011). Because SNP1 was a non-synonymous change, it altered the deduced amino acid sequence of the SERF protein from alanine to valine at position 64 in LU132 (Fig. 4A). The amino acid change had the potential to change the protein structure. Phyre2 and SAM-T08 predicted four helices for the LU132 and three helices for the LU140 protein (Fig. S1) (Karplus, 2009; Karplus et al., 2005; Kelley & Sternberg, 2009). ELM identified a mitogen-activated protein kinase docking motif (DOC_MAPK_1) with the amino acid pattern KKRxxKxxxxLxV created by SNP1 in LU132 but absent in LU140 (Dinkel et al., 2013) that could potentially change the function of SERF.

Figure 4 Confirmed annotation of SNP-containing serf and pcna.

The gene models were corrected compared to those of the reference based on cDNA sequences. (A) The serf gene was 21 bp longer than predicted by the IMI206040 annotation, contained three exons and SNP1 resided at bp position 191 in the 3rd exon of serf. SNP1 changed the amino acid sequence of SERF in LU132 compared to other strains. (B) The sequenced part of the pcna transcript was 3 bp shorter than the IMI206040 annotation as a result of different intron-exon boundaries. SNP2 was a synonymous change in pcna in LU140 compared to other strains. aa, amino acid.

SNP2 was located in the third exon of a gene (protein ID 212486: GenBank accession number XP_013948189.1) containing a putative conserved vezatin superfamily domain (pfam12632). Vezatin is a peroxisome transmembrane receptor that is involved in membrane-membrane and cell–cell adhesions (Marchler-Bauer et al., 2011). The protein had low level similarities (50–60%) to proliferating cell nuclear antigen (pcna) from Beauveria bassiana ( XP_008593672), Colletotrichum orbiculare (ENH88404) and Togninia minima ( XP_007918421), but its function in these organisms was not known. SNP2 was located in the 3rd exon of pcna but was a synonymous change that did not alter the deduced amino acid sequence for the PCNA protein (Fig. 4B).

The identified two SNP-related target genes (serf and pcna) were selected for further analyses to determine whether their gene expression or functions were likely to be affected by the SNPs.

Relative expression of target genes

The relative expression of the two SNP-related target genes (serf and pcna) and the two metabolism-related target genes (genes encoding N-acetyl-β-D-glucosaminidases nag1 and nag2) was assessed under standard culturing and mycoparasitism-inducing conditions. Gene expression, normalised to the expression of the reference gene (tef1α), were similar in LU132 and LU140 (Table S7). Induction with N-acetyl-D-glucosamine (NAG) resulted in significantly higher relative expression than without NAG of nag1, nag2 and pcna in LU132 and of nag2 and pcna in LU140. The results of this experiment showed that the different phenotypes of LU132 and LU140 were not caused by differential expression of the two SNP-harbouring genes or the two NAGase-encoding genes.

Functional analysis of serf

Functional analysis of the serf gene was carried out by gene replacement in LU132 and LU140 with the aim of generating identical mutant strains. Three monokaryotic mitotically stable hygromycin B resistant Δserf transformants for LU132 (mutants A, B and C) and LU140 (mutants D, E and F) were generated. PCR and Southern analysis confirmed that both LU132 WT and LU140 WT contained a single copy of serf at the predicted position in the genome and that serf was replaced by a single copy of the knock-out construct in all mutants (Figs. S2 and S3). The average growth rates (grand mean) of the LU140 Δserf mutants were not significantly different from LU140 WT (P < 0.05) (Table 2). The average growth rates of the LU132 Δserf mutants varied significantly from the LU132 WT in all treatments. Two LU132 Δserf mutants (B and C) displayed a growth rate reduction of 3% while one mutant (A) had a reduction of 9% compared to the LU132 WT. However, even the slowest growing LU132 mutant (A) grew on average significantly faster than LU140 WT.

The colony appearances of the mutants that were incubated with constant light, are shown in Fig. 5. On PDA (pH 5) the conidia of LU140 and its mutants D, E and F and the LU132 mutant A covered the whole plate, while LU132 and its mutants B and C did not produce conidia in the centre. On PDA pH 2.7, LU140 and its mutants D, E, F and the LU132 mutant A all produced yellow-green conidia while LU132 and its mutants B and C produced only immature white/light green conidia.

Figure 5 Colony appearance/conidiation of mutants.

LU132 mutants B and C always resembled the LU132 WT, while LU132 mutant A resembled the LU140 WT. The conidiation characteristics of all LU140 mutants were similar to the 140 WT.

The phenotype microarray assay (Biolog FF) was used to group the mutants according to their metabolic profiles (Fig. S4). The cluster analysis of the OD750 data (mycelial growth) resulted in two main groups. One group contained LU132 WT and its Δserf mutants B and C while the second group consisted of LU140 WT, all its mutants (D, E and F) and LU132 mutant A. The two groups separated at a similarity distance of 0.49 (a value of 1 represents complete similarity and 0 complete dissimilarity). Analysis of the conidiation resulted in the same two groups that separated at a similarity distance of 0.54. The OD490 data (catabolic activity) were not so clearly grouped. One strain (LU132 mutant C) was an outlier and the remaining strains were separated into two groups at a similarity distance of 0.67. One group contained LU132, its mutant B and the two LU140 mutants E and F, while the second group contained LU140, its mutant D and LU132 mutant A.

In summary these data show that the deletion of serf in LU132 and LU140 did not result in identical phenotypes. Despite the fact that SNP1 in serf was the only detectable functional genomic difference between LU132 and LU140, functional analysis with six independent Δserf mutants suggested that this only partially accounted for the phenotypic differences.

Discussion

The main aim of this study was to identify genetic differences between genetically highly similar T. cf. atroviride strains LU132 and LU140 by whole genome comparison. This research allowed the identification of two SNPs that distinguish the strains.

One SNP could be utilised for the successful development and validation of a molecular test for the commercial biocontrol strain T. cf. atroviride LU132. The marker screening technique is straightforward and affordable. The availability of the marker will enable the registration of LU132 biocontrol products internationally, thus opening them to additional markets. The molecular marker for LU132 will also enable more detailed research on this particular strain, as it will now be possible to identify this strain in the environment to study colonisation, competition and mycoparasitism.

The genomes of LU132 and LU140 were found to be nearly identical. Earlier molecular studies with the two strains indicated that there would only be a small number of genomic differences, but the identification of only 1 SNP per strain was very surprising. It has to be noted though that the inability to identify a SNP is not proof of absence and that SNPs in repetitive regions or in regions with missing coverage may have been missed in this exercise. However, Trichoderma atroviride contains only a small number of degenerate transposable elements (0.49% of the genome) and micro- or mini satellite DNA (0.94%) (Kubicek et al., 2011). The analysis of de novo assembled unmapped reads for LU132 and LU140 minimised the possibility that SNPs might have been missed because they were located in genomic regions with low coverage. In addition to that, the genomes of two other T. cf. atroviride strains from New Zealand were analysed at the same time, using the same methods, and were found to have on the order of 10,000 times more strain-specific SNPs than LU132 and LU140 (data not shown), confirming the appropriateness of the applied methods to identify SNPs. In fact, simple by-chance scrolling through these sequences in IGV enabled the identification of a SNP and a 1 bp insertion in these other strains (Figs. S5 and S6).

The revelation of the high sequence identity of LU132 and LU140 made it necessary to confirm the strains’ individualities. The phenotype assays confirmed the strains to be individuals, exhibiting distinct growth rates, conidiation patterns and metabolism. The mode of action that makes LU132 a more successful BCA than LU140 under laboratory conditions appears to be its faster growth and pH adaptability. These attributes would make LU132 an especially successful competitor. Competition for nutrients has earlier been identified as one way in which LU132 controls Botrytis cinerea on strawberries (Card et al., 2009). The greater pathogen inhibition on dual culture plates by LU132 can be attributed to its faster growth alone. As no inhibition zones were formed between Trichoderma strains and pathogens, the inhibition was probably not caused by secreted antibiotics.

The metabolism and conidiation profiles of LU132 and LU140 could be distinguished using Phenotype Microarrays. N-acetyl-D-glucosamine (NAG) was one of the compounds that resulted in significantly different growth and catabolic activity of LU132 and LU140. NAG is a monomer of chitin, the main component of fungal cell walls. Trichoderma spp. contain two NAG-cleaving enzymes (N-acetyl-β-D-glucosaminidase 1 and 2) encoded by nag1 and nag2 (Mach et al., 1999; Peterbauer et al., 1996; Ramot et al., 2004). These enzymes are not only involved in chitin degradation of fungal cell walls (Brunner et al., 2003) and in mycoparasitism (Zeilinger et al., 1999) but also in mycelial growth on chitin (Lopez-Mondejar et al., 2009). However, LU132 and LU140 expressed these two genes at similar levels. By comparing NAGases activity to transcript levels of nag1 and nag2 in T. atroviride P1 on Biolog FF plates, Seidl, Druzhinina & Kubicek (2006) found that both genes are regulated at the transcriptional level. This suggests that the similar expression levels of nag1 and nag2 in LU132 and LU140 would lead to similar NAGase activities in both strains and that their different growth rates are not likely to be affected by NAG metabolism. The difference in the biocontrol abilities of the two strains appears therefore to be more complex than originally assumed.

The two identified SNPs did not alter the expression of the associated target genes; however, SNP1 in serf was a non-synonymous change. Bioinformatics analyses predicted protein structure changes due to the amino acid change in SERF from LU132, which could have impacts on the protein function. A single change in the amino acid sequence could therefore potentially have effects on multiple processes (pleiotropy) and might lead to the altered phenotype of LU132, compared to LU140, in multiple complex ways. The more efficient biocontrol activity of LU132, compared to LU140, might therefore be a result of multiple changes caused by SNP1 rather than a result of SNP1 directly. This hypothesis correlates with the finding that a number of metabolic differences were found between LU132 and LU140 but the main biocontrol-related phenotypic difference was found in the growth rates.

To study the involvement of a particular gene in the development of a mutant phenotype, the mutation is usually introduced into the wild type to achieve targeted gene disruption, and the mutated gene then replaced with the wild type gene in the mutant. These processes include the introduction of selection markers that can also have phenotypic effects. For the analysis of a particular gene function this is generally not a problem. In contrast to this, our aim was to create genomically identical strains to see if their phenotypes would be identical. The only current option to achieve this was to remove the SNP-containing gene in both strains and to replace it with an identical knockout cassette, including an identical selection marker.

The deletion of serf in LU132 and LU140 did not result in identical mutant phenotypes, implying that SNP1 was not the only reason for the phenotypic differences between LU132 and LU140. Even though the change in serf was the only apparent functional genetic change, it did not entirely account for the phenotypic differences. This result indicates that multiple factors must be involved in the development of the dissimilar phenotypes. As the expected phenotypes (LU132 and LU140 being identical) could not be generated by deletion of serf, a complementation was not attempted. Reinstating the gene might have confirmed the involvement of serf in the development of the observed mutant phenotypes; however, it would not have provided more information about the additional factors responsible for the different wild type phenotypes.

Epigenetic modification could contribute to development of the distinct phenotypes of LU132 and LU140. Interactions between pathogen, plant and the biocontrol agent are very complex and are therefore difficult to associate with a genetic origin. This is attributed to multiple genetic causes but also to epigenetic modifications, such as DNA methylation, histone modifications and RNA interference. It has been shown for instance that the DNA methylation states of three dimorphic fungi (M. rouxii, Y. lipolytica and U. maydis) differ between their mycelial and yeast stages (Reyna-Lopez, Simpson & Ruiz-Herrera, 1997), that chromatin-remodelling and DNA methylation affect gene expression in Neurospora (Belden et al., 2011), that histone modifications lead to transcriptional activation or repression in many fungi (Aghcheh & Kubicek, 2015) and that non-coding micro-RNA like RNAs (milRNAs) could be potential regulators of cellulase production or fungal growth in T. reesei (Kang et al., 2013). Genome comparisons showed that Trichoderma atroviride, T. virens and T. reesei contain genes or homologs to the genes known to be involved in epigenetic regulation of gene expression (Schmoll et al., in press) but the actual functionality of these processes have not yet been studied in Trichoderma.

Another reason for the different phenotypes of LU132 and LU140 could be that one or both strains naturally contained an extra-chromosomal element, such as a mycovirus or a plasmid. Mycoviruses are widespread in fungi (Ghabrial & Suzuki, 2009) where they can affect virulence and cause debilitation (McCabe, Pfeiffer & Van Alfen, 1999; Preisig et al., 2000). Although the function has not yet been studied, single-stranded RNA elements could be identified in one other T. cf. atroviride strain from New Zealand but no extra-chromosomal elements have been found in LU132 and LU140 (Lange, 2015). To our knowledge, only double-stranded RNA (dsRNA) elements have been reported for other Trichoderma species so far and their impact on the phenotype is equally unknown (Antal et al., 2005a; Antal et al., 2005b; Jom-in & Akarapisan, 2009). Circular plasmids have been identified in mitochondria of Trichoderma viride, T. harzianum and T. virens (Antal et al., 2002; Meyer, 1991). The plasmids appeared to have no influence on the strain’s morphology; however, it is known that a plasmid in Neurospora species is responsible for senescence (Griffiths, Kraus & Bertrand, 1986).

Epigenetic modification or extra-chromosomal elements could also explain the phenotypic differences between the three LU132 Δserf mutants. The phenotypic differences were intriguing, as all mutants were confirmed to have identical sequences in the manipulated genomic region and all contained exactly one copy of the knock-out cassette. LU132 might naturally harbour an extra-chromosomal element, that only got lost from mutant A during the mutagenesis, with the result that A’s phenotype was more similar to LU140’s phenotype. Epigenetic modifications would not necessarily have been changed by the mutation process and could therefore have led to the observed variable results.

Conclusion

The main goal to iden tify genetic differences between genetically highly similar T. cf. atroviride strains LU132 and LU140 by whole genome comparison was successful. A strain-specific molecular marker for T. cf. atroviride LU132 was successfully designed and validated. Further analysis of the polymorphic gene, containing the non-synonymous SNP1, highlighted that even apparently genetically identical strains (Δserf mutants) can have different phenotypes and that natural strains with different phenotypes (LU132 and LU140) can be genetically extremely similar. Even though whole genome sequencing is an important tool for fundamental and applied research, the definition of an individual is not exclusively defined by its DNA sequence. In the microbiological context, this creates limitations for molecular strain typing to identify efficient biocontrol strains or pathogens and implies that these techniques should not be applied in isolation but should always be combined with phenotypic characterisation.

Data Availability

Strains are available upon request. Gene sequence data are available at GenBank accession numbers: KR812141.1 (serf for LU132), KR812142.1 (serf for LU140), KR812145.1 (pcna for LU132), KR812146.1 (pcna for LU140) and EHK42777.1 (tef1α). Illumina raw data for LU132 and LU140 are available at the NCBI sequence read archive, accession SRP070858.

Supplemental Information

Figure S1 Tertiary structure prediction for SERF from LU132 and LU140

(A) Phyre2 prediction, blue, N terminus; red, C terminus; four α-helices for LU132 and three for LU140. The approximate location of the amino acid change is indicated by “SNP1”. (B) Sam-T08 prediction, red for α-helix and grey for other, four α-helices for LU132 and three for LU140.

Click here for additional data file.

Figure S2 PCR confirmation of mutants

(LF) Left flank of SKO was amplified with primers A/C (Table S2). The 1.6 kb band was present in all mutants (A–F) and absent in the WT. (RF) Right flank of SKO was amplified with primers D/B. The 1.4 kb band was present in all mutants (A–F) and absent in the WT. (serf) serf was amplified with primers SNP1-F/B. The 1.4 kb band was absent in mutants A, C, D, E and F and present in mutant B and in the WT. (SKO) The whole SKO construct was amplified with primers A/B. The 3.6 kb band was present in all mutants and the 2.6 kb band was present in the WT. (serf- repeat) After another purification round for mutant B, no serf DNA could be detected anymore. Arrows indicate the above mentioned bands, WT DNA was used as positive control and size standard was the 1 Kb Plus DNA Ladder™ (Invitrogen).

Click here for additional data file.

Figure 3 Southern blot confirmation of mutants

(A) The SacI and PstI digested Δserf mutant DNA was hybridised with the hph probe. A pki/hph containing T. atroviride IMI206040 Δblr-2 mutant was used as positive control. (B) The SacI and PstI digested WT DNA was hybridised with the serf probe. Both hybridisations resulted in single bands, confirming the presence of single copies. Unlabelled probes were used as positive controls and size standard was the 1 Kb Plus DNA Ladder™ (Invitrogen).

Click here for additional data file.

Figure S4 Cluster analysis of metabolic profile data

The eight strains were clearly separated into two groups regarding their mycelial growth (OD750) and conidiation on 95 different nutrient sources. The OD490 data were more homogeneous, resulting in higher similarity distances at the branching nodes of the dendrogram.

Click here for additional data file.

Figure S5 IGV screenshot of a SNP in T. cf. atroviride LU633

Simple scrolling through the genome sequences enabled the by-chance identification of a SNP (green) in LU633 compared to LU132, LU140, LU584 and T. atroviride IMI206040 (on the bottom panel).

Click here for additional data file.

Figure S6 IGV screenshot of an insertion in T. cf. atroviride LU584

Simple scrolling through the genome sequences enabled the by-chance identification of an insertion (purple) in LU584 compared to LU132, LU140, LU633 and T. atroviride IMI206040 (on the bottom panel).

Click here for additional data file.

Table S1 Colony morphology—experimental design

∗ Colonies derived from conidia suspensions, † colonies derived from agar plug containing colony arising from single conidium, ub unbuffered, b buffered, ‡ the conidial yield was assessed for one treatment in Exp 3, the final pH of the buffered and unbuffered PDA in Exp 3 and 4 was measured using a flat tip pH probe.

Click here for additional data file.

Table S2 Primers

Underlined are additional sequences to introduce restriction sites, ∗ Introducing sfiI restriction sites, † previously designed within our group.

Click here for additional data file.

Table S3 Average growth rates (mm/d) of LU132 and LU140

Different letters represent significantly different values (P < 0.05) for each experiment or for each row (as indicated). The biggest differences between isolates is highlighted in bold. ∗ The actual pH of the plates was measured before inoculation using a flat tip pH probe.

Click here for additional data file.

Table S4 Pathogen growth in mm and inhibition (%) by LU132 and LU140 on dual culture plates

∗ The same pathogen in place of the antagonist acted as control. With P. ultimum the inhibition is negative because the pathogen grew further on the treatment plate than on the control plate. The pathogen inhibition is displayed as % of pathogen growth reduction on treatment plate in relation to control plate. Different letters represent significantly different values (l.s.d. = 1.559) (P < 0.05).

Click here for additional data file.

Table S5 Conidiation scores and number of wells with conidia

∗ Average conidiation score of 95 wells of three Biolog FF plates. † Number of wells that contained conidia (or pustules). Different letters represent significantly different values (l.s.d. = 0.427 ) (P < 0.05).

Click here for additional data file.

Table S6 Trichoderma strains used for marker validation

Click here for additional data file.

Table S7 Relative expression of target genes

Lower ΔCqs mean higher relative expression and vice versa. Different letters within rows represent significantly different values (P < 0.05).

Click here for additional data file.

The pki/hph and Δblr-2 fragments (derived from plasmid pCB1004 (FGSC)) were kindly provided by Artemio Mendoza-Mendoza (Bio-Protection Research Centre, Lincoln University, Lincoln, New Zealand), pYT6 was kindly provided by Barry Scott (Massey University, Palmerston North, New Zealand) and Pythium ultimum spp. was kindly provided by Wadia Kandula (Bio-Protection Research Centre, Lincoln University, Lincoln, New Zealand). We wish to thank Andrew Holyoake for theoretical and technical support as well as Dave Saville for help with statistical analyses.

Additional Information and Declarations

Competing Interests

Author Contributions

DNA Deposition

Data Availability

Alison Stewart is an employee of Scion, Rotorua, New Zealand.

Claudia Lange conceived and designed the experiments, performed the experiments, analyzed the data, contributed reagents/materials/analysis tools, wrote the paper, prepared figures and/or tables, reviewed drafts of the paper.

Richard J. Weld, Alison Stewart and Johanna M. Steyaert conceived and designed the experiments, contributed reagents/materials/analysis tools, reviewed drafts of the paper.

Murray P. Cox and Kirstin L. McLean conceived and designed the experiments, performed the experiments, analyzed the data, contributed reagents/materials/analysis tools, reviewed drafts of the paper.

Rosie E. Bradshaw conceived and designed the experiments, reviewed drafts of the paper.

The following information was supplied regarding the deposition of DNA sequences:

GenBank: KR812141.1, KR812142.1, KR812145.1, KR812146.1, EHK42777.1.

The following information was supplied regarding data availability:

NCBI SRA: SRP070858.

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
