# Peer review of "Genome-scale investigation of phenotypically distinct but nearly clonal Trichoderma strains"

_PeerJ, doi:10.7717/peerj.2023_

## Round 0.1 · original submission · Minor Revisions

Although all reviewers appreciated the study and found the findings interesting, two suggested revisions that should be made to the text and writing (minor revisions).

Reviewer 1 ·

Basic reporting

The manuscript is easy to follow and written in professional English language.
The introduction is well referenced with relevant citations, however, I think that Braithwaite et al. (2016) appearing as "forthcoming" is not an acceptable reference, at least the in press stage should be accessible, otherwise the format „Braithwaite M et al., unpublished” should be used in the text. Furthermore, accessibility via full text links should be provided for references citing the thesis works of Dodd-Wilson SL, Kay SJ, Lange C, McLean K 1996 and McLean K 2001.
FigS4: should be „data represent” (plural)
I missed conventional legends in the case of the supplementary figures.

Experimental design

The manuscript reports original primary research within the scope of the journal. The research questions are well defined, the experimental design and the applied microbiological, biochemical, genome-scale, bioinformatical and statistical tools are appropriate. The authors performed a remarkable amount of work at a high technical standard. The methods are described with sufficient detail and information to repeat the experiments.

Validity of the findings

Genomic sequence data are made available in the NCBI sequence read archive database.

Additional comments

The manuscript of Lange et al. reports the morphological, biochemical and genome-scale comparison of two nearly clonal strains of Trichoderma cf. atroviride. The authors describe the clearly different phenotypic characteristics and antagonistic behaviour of the two strains which they try to relate to differences at the genomic level. Remarkably, the full genome comparison of the two strains revealed only 2 SNPs within coding regions of genes, with only one of them being non-synonymous, which made the explanation of the phenotypical differences between the two strains with genetic background difficult, but enabled the development of a strain-specific monitoring tool for the biocontrol strain LU132.
The results represent an interesting contribution to both fungal comparative genomics and biocontrol research.

Question: In section 175-182 the authors report about the identification of 659 putative SNPs, and finally state that two SNPs, both transition polymorphisms could be confirmed. Does it mean that all other putative SNPs were false positives? Were there any polymorphisms between the two strains in non-coding genomic regions? If yes, could this provide an explanation for the different phenotypic characters?

Minor comments:
71: SCAR is not a specific „region” but a strategy, see e.g. Naeimi et al: Strain-specific SCAR markers for the detection of Trichoderma harzianum AS12-2, a biological control agent against Rhizoctonia solani, the causal agent of rice sheath blight. ACTA BIOLOGICA HUNGARICA 62: 73-84. (2011)
79: „more than 200 species” would be better
89, 92: „hypercellulolytic” would be more appropriate instead of „cellulase producing”, biocontrol strains may also produce cellulases, even if in smalller quantities
Figure 2: the greek letter should be used for alpha instead of „a” in the name of alpha-D-glucose
276 – Table 2: it is not defined, what the data in the table are. Are they radial growth rates in mm/day?
168 and 501: link is not working (03.21.2016)
356: should be „Seidl, Druzhinina & Kubicek (2006)”
487: should be „Bochner, Gadzinski & Panomitros 2001” as in line 484
492: should be „Friedl, Kubicek and Druzhinina 2008” a sin line 141
514: should be „BLAST analysis”
539: should be „Seidl, Druzhinina & Kubicek 2006” as in line 154
555: should be „from Holyoake et al. (2008)”
620: should be „Mukherjee M” instead of „Mala M”
623: journal name is abbreviated here but not elsewhere – follow the required format of the journal
647: „GR S, editor” ?
665, 710, 761, 763: missing dot and/or space at the end of title
679, 744 and 809: „FEMS”, „FEBS” and „SGM” should be in all caps
758, 759: „Microbiology”, „Journal of Biological Chemistry” with capital initial
803-804: do not capitalize title initials
References: doi numbers are indicated for certain references but not for others – follow the required format of the journal
810-811: the bibliographic data are missing from reference Shoukouhi P, and Bissett J. 2009
Tables S1, S3 and S5: data should be given with decimal points, not commas
Table S7: data are missing for primers C, D and SNP1-F

Reviewer 2 ·

Basic reporting

no comments

Experimental design

p. 10, starting with line 183, describes de novo assembly of the reads that did not map to the reference T. atroviride genome, but there is no mention of regions of the reference of the genome that had no coverage. These are not necessarily included in the assembly of the unmapped reads. Was there any effort to test whether regions lacking coverage are indeed missing from the LU132 and/or LU140 genomes (by individual PCR reactions for example)? It seems this might have identified addtional differences. Probably I am missing something, if so please clarify in the text.

Validity of the findings

The conclusions are well-supported by the data.

Additional comments

The key point of this paper is that two very similar strains of Trichoderma atroviride are very different in their biocontrol ability. This provided the opportunity to use genome sequencing to identify the genetic basis for the different phenotypes of the two strains. The difference that was identified accounted, in part (but not completely) for the phenotypic difference. An additional outcome was the development of the SNPs as markers for identification of the LU132 strain in the field, with immediate practical application.

·

Basic reporting

I believe that this manuscript is appropriated for publication in PeerJ.

It is well written and edited, recent and relevant bibliography has been used and figures and tables are clear.

Experimental design

The objective of this manuscript is clear and original.
Methods have been described with datail.
I think it's a very complete job with many well-designed experiments and it can contribute to a better understanding of Trichoderma at a molecular level.

Validity of the findings

Results are robust and interesting, although they could be summarized.
Discussion is good, remarkable conclusions and identified speculations can be drawn.

---

## Round 0.2 · accepted · Accept

All reviewer comments were adequately addressed.